# Mechanism of Decision Making between Autophagy and Apoptosis Induction upon Endoplasmic Reticulum Stress

**DOI:** 10.3390/ijms25084368

**Published:** 2024-04-15

**Authors:** Orsolya Kapuy

**Affiliations:** Department of Molecular Biology, Institute of Biochemistry and Molecular Biology, Semmelweis University, H-1085 Budapest, Hungary; kapuy.orsolya@semmelweis.hu

**Keywords:** endoplasmic reticulum stress, unfolded protein response, autophagy, apoptosis, bistability, systems biology

## Abstract

Dynamic regulation of the cellular proteome is mainly controlled in the endoplasmic reticulum (ER). Accumulation of misfolded proteins due to ER stress leads to the activation of unfolded protein response (UPR). The primary role of UPR is to reduce the bulk of damages and try to drive back the system to the former or a new homeostatic state by autophagy, while an excessive level of stress results in apoptosis. It has already been proven that the proper order and characteristic features of both surviving and self-killing mechanisms are controlled by negative and positive feedback loops, respectively. The new results suggest that these feedback loops are found not only within but also between branches of the UPR, fine-tuning the response to ER stress. In this review, we summarize the recent knowledge of the dynamical characteristic of endoplasmic reticulum stress response mechanism by using both theoretical and molecular biological techniques. In addition, this review pays special attention to describing the mechanism of action of the dynamical features of the feedback loops controlling cellular life-and-death decision upon ER stress. Since ER stress appears in diseases that are common worldwide, a more detailed understanding of the behaviour of the stress response is of medical importance.

## 1. Introduction

Endoplasmic reticulum (ER) is a continuous membrane system of each eukaryotic cell forming a series of flattened sacs within the cytoplasm [1,2]. ER has a crucial role in sensing cellular homeostasis and generating suitable signals and responses upon external and internal stimuli [3,4,5]. Depending on whether ribosomes are attached to the surface of the ER, we distinguish smooth endoplasmic reticulum (SER) and rough endoplasmic reticulum (RER), respectively [6]. SER and RER have several differences in certain physical and functional characteristics of ER [7].

SER has a key role in cellular metabolism (such as lipid biosynthesis and carbohydrate metabolism) and several signalling processes as well [8,9]. For these integrated roles of ER, a special redox homeostasis and a high luminal Ca^2+^ environment are required [2,10,11].

RER is specialized in the synthesis of those proteins which have major functions in synthesizing, folding, packaging and transporting secreted and membrane proteins of the cell [12,13]. Proteins synthesized in the RER are either embedded in the ER membrane or are translocated into the RER lumen via a translocon channel [14]. These proteins take their final form in the lumen of the ER (called protein folding) and can undergo various post-translational modifications (such as glycosylation) inside the ER [15,16,17]. The RER is located near the Golgi apparatus, another eukaryotic organelle, which has an essential role in transporting, modifying and packaging proteins for delivery to targeted destinations [2,15]. Unnecessary, damaged or improperly folded proteins are directly transported from the Golgi apparatus to the lysosome for degradation [16,18,19]. Perturbation of ER homeostasis can result in the overwhelming of folding capacity of ER, causing an effect on the cell called ER stress. Several external and integral negative stimuli can compromise the homeostasis of ER inducing ER stress, such as nutrient deprivation, hypoxia and calcium depletion [1,2]. ER plays a crucial role in cellular homeostasis by sensing and generating signals to drive cellular responses [3]. Due to the complex role of ER in the normal physiological functions of the cell, dysfunctional behaviour of ER might have serious consequences [2,3,10,20]. ER stress is implicated in a wide range of human diseases [21,22], such as diabetes [23], non-alcoholic fatty acid disease [24], inflammations (including heart diseases [25] and inflammatory bowel disease [26,27]), neurodegeneration (Parkinson’s disease [28], Alzheimer’s disease [29] and Huntington’s disease [30,31]) and also psychiatric diseases [32,33]. Since defects in the proper functioning of the ER are observed in most human diseases and the frequency of ER stress is also increased during aging [34,35,36], it is crucial to study and understand the dynamic behaviour of the regulatory system as comprehensively as possible.

In this review, we summarize the recent knowledge of the dynamical characteristic of endoplasmic-reticulum-stress-induced response mechanisms by using both theoretical and molecular biological techniques. We investigate how the cellular decision is made of choosing between autophagy-dependent survival and apoptotic cell death. This review pays special attention to describing the dynamical roles of the positive and negative feedback loops controlling cellular life-and-death decision upon ER stress.

## 2. ER Stress Can Induce Both Autophagy and Apoptosis

The ER is a eukaryotic cellular component that acts as an essential integrator of external and internal stimuli by keeping the proper balance of protein levels (so called proteostasis) [3,11]. Since secreted and membrane proteins are folded and matured in the ER lumen, and then, they are transferred and displayed on the cell surface or released extracellularly, a precise quality-control equipment of the ER is essential to ensure cellular protein homeostasis [11,37]. The precise balance between production and consumption of folded proteins is tightly controlled, while accumulation of incorrectly folded proteins in ER lumen leads to harmful ER stress [3,38]. The primary purpose of the cell is to compensate for these effects and ensure cellular survival, but unresolvable or persistent adverse effects lead to cell death [39]. Several scientific results have confirmed that macroautophagy (here called autophagy) plays an essential role in cell survival after ER stress by self-digesting the damaged components [40,41]. However, an excessive level of ER stress results in apoptotic cell death [41,42,43].

Traditionally, autophagy is a type of programmed cell death mechanism, where the damaged or unnecessary cellular components are self-digested by an evolutionary conserved process [44,45]. Cells always have some basal autophagic activity even under physiological conditions; however, the process becomes more efficient at various stress events (i.e., starvation, ER stress) [44,46]. During autophagy, cellular components become sequestered into a double membrane vesicle called autophagosome [47]. This autophagosome can fuse with the lytic enzymes containing lysosome to achieve a complete decomposition of the secreted components [45,48]. Due to the crucial role of autophagy in maintaining cellular homeostasis, this self-eating process is precisely regulated [47,48]. Autophagy is essentially a reversible process: once the damaged components have been successfully digested, autophagy is switched off [49]. Interestingly, both the absence of autophagy and an excessive level of autophagy are able to cause cell death [50].

The main role of apoptosis is to eliminate aberrant or seriously damaged cells, but it also has an important role in removing cells upon embryonic development and maturation of the immune system [51,52,53]. During apoptosis, the selected cells can be abolished in a controlled way by cellular shrinkage, mitochondrial permeabilization and DNA fragmentation [54,55]. Apoptotic cell death can be induced by two pathways, called extrinsic and intrinsic pathways, respectively [56]. While the mitochondria-located intrinsic pathway can be activated by the wide range of cellular stress signals (such as DNA damage, ER stress), the extrinsic pathway is initiated by death-receptors [55,57]. Apoptosis should always be an irreversible process; once the cell has decided to commit suicide, it can never return [53,58]. That is why apoptosis must not be triggered by weak signals, and the cell must not hesitate on the borderline of non-apoptotic and apoptotic state; otherwise, it can have serious consequences for the cellular system [59].

## 3. The Characteristic of Endoplasmic Stress Response Mechanism

The precise balance between production and consumption of folded proteins and the accumulation of unfolded or misfolded proteins are tightly regulated by an evolutionarily conserved complex network of signalling pathways called unfolded protein response (UPR) [60,61,62]. Since the accumulation of incorrectly folded proteins in ER lumen leads to harmful ER stress, the primary role of UPR is to avoid cell damage in response to ER stress [22].

### 3.1. The Molecular Mechanism of Unfolded Protein Response

The signalling pathway of UPR has three well-defined transducers activated by ER stress, called IRE1 (inositol requiring 1), PERK (protein kinase RNA-like endoplasmic reticulum kinase) and ATF6 (activating transcription factor 6), respectively [60,62] (Figure 1). All three components are ER-resident transmembrane proteins, which are bound to the luminal domain of GRP78 (glucose-regulated protein), also known as BiP (binding immunoglobin protein) under physiological conditions [63,64]. Misfolded or unfolded proteins that accumulate during ER stress attract chaperone proteins 64that help them to coil up properly, such as BiP, thus releasing the three main inducers of the UPR [65,66]. Subsequently, PERK and IRE1 become activated by multimerization and trans-autophosphorylation, while ATF6 is translocated to the Golgi apparatus where it is proteolytically processed into the cytoplasm-soluble and active transcription factor ATF6f (p50) [62,63,67].

PERK−/− cells are hypersensitive to the lethal effect of ER stress, suggesting their essential role in the stress-response mechanism [68,69]. The active PERK is able to phosphorylate the translation initiation factor eiF2α (eukaryotic translation initiation factor 2α), which becomes a potent allosteric inhibitor of eIF2B [70]. Therefore, eiF2α phosphorylation reduces the global protein synthesis, decreasing the flux of protein entering the ER [70]. Phosphorylated eiF2α enhances the initiation of ATF4 (activating transcription factor 4) mRNA, resulting in the upregulation of ATF4 protein [71]. Two downstream targets of ATF4 are GADD34 (growth arrest and DNA damage-inducible 34) [72] and CHOP (transcription factor C/EBP homologues protein) [68], respectively. IRE1 can induce its RNase activity resulting in the initiation of the unconventional splicing of a transcription factor, known as the spliced X-box binding protein 1 (sXBP1), and it also promotes the cJUN N-terminal kinase (JNK) signalling pathway upon ER stress [61,73,74]. The activation of both IRE1 and ATF6 promotes cell survival by positively regulating the transcription of several genes involved in protein folding, protein quality control, phospholipid synthesis and ER-associated degradation (ERAD) [1]. 

### 3.2. The UPR Dependent Regulation of Both Autophagy and Apoptosis

All three pathways of UPR play a key role in the cellular life-or-death decision under ER stress by choosing between autophagy-dependent self-cannibalism and apoptotic cell death, according to the level of ER stress [39] (Figure 1).

The inhibition of GADD34 getsdiminished in autophagy-dependent survival suggesting that GADD34 has a positive effect on autophagy induction [75,76,77]. Additionally, it is able to down-regulate apoptotic cell death via mTOR inhibition during ER stress [76,78]. Recently, three classes of autophagy genes have been identified according to their dependence on ATF4 and CHOP and the binding of these factors to the promoter [3,79,80]. B‘chir et al. have distinguished autophagy genes whose induction depends only on ATF4 (e.g., Atg16L1, Atg12, Beclin1), only on CHOP (e.g., Atg10, Atg5) or on both ATF4 and CHOP (e.g., p62, Atg7), suggesting that both ATF4 and CHOP play a direct role in autophagy-dependent survival under ER stress [80]. Interestingly, either CHOP or GADD34 overexpression reduce cell viability and result in apoptotic cell death upon cellular stress [81,82]. CHOP is a transcription factor that controls gene transcription involved in apoptosis [83,84]. CHOP promotes the down-regulation of Bcl-2 and the induction of the BH3-only pro-apoptotic proteins Bim, Puma and Bax as well as DR5, a member of the death-receptor protein family, to induce the self-killing mechanism [83,84,85,86]. In addition, CHOP-deleted cells are much less sensitive to ER stress compared to wild type strain [87,88].

Since the presence of sXBP1 can induce autophagy vesicle formation, the XBP1 deficiency abolishes the autophagy response upon ER stress [3,89]. The XBP1-dependent transcriptional upregulation of many autophagy receptor genes (such as p62, LC3, Beclin1) is observed during ER stress [90]. It has been also shown that transient overexpression of sXBP1 induces the synthesis of various autophagy markers and promotes proliferation in bone-marrow-derived macrophages, while an excessive level of XBP1s leads to apoptotic cell death [91,92]. Interestingly, the ER-stress-induced IRE1-JNK pathway is able to promote autophagy-dependent survival in the initial phase of hepatic steatosis, while inhibition of the pathway results in the diminishing of autophagy and an increase in apoptosis [93]. JNK pathway also down-regulates B-cell CLL/lymphoma 2 (Bcl2) upon ER stress, inducing the release of Beclin1, the central activator of autophagy [94,95,96]. However, it is also well known that the JNK signalling pathway is essential for apoptotic cell death via induction of both the regulation Bcl2 family of proteins and the caspase cascade [61,73]. 

ATF6 increases the expression of death-associated protein kinase 1 (DAPK1), which has an essential role in the Beclin1-dependent autophagy induction [97,98]. Additionally, ATF6 promotes apoptotic cell death with the reduction in anti-apoptotic proteins upon ER [25,99]. In addition, overexpression of active ATF6 induces apoptosis in myoblast cells [100].

## 4. Systems Biological Analysis of Autophagy and Apoptosis Induction upon ER Stress

Kapuy et al. have recently shown that the key autophagy and apoptosis inducers mutually inhibit each other, forming a double negative feedback loop in the control network (e.g., Beclin1 inhibits Caspase3, while the active Caspase3 induces the cleavage of Beclin1) upon cellular stress (such as ER stress) [59,101]. Mutual antagonism between the survival and the self-killing mechanisms means that they are mutually exclusive: they can never be active at the same time in the event of stress induction [102,103,104]. The question immediately arises, of how the control network chooses between the two mechanisms, if UPR is able to induce both autophagy and apoptosis upon ER stress.

### 4.1. The Dynamical Analysis of Autophagy and Apoptosis Induction upon ER Stress

To explore the dynamical characteristics of the control network, recently, a systems biology study was carried out where both theoretical and experimental techniques were used [105,106]. First, a minimal model was built up (Figure 2a), which was independent of the identity of the molecular players and only includes the main system level feedback loops to plot the balance curves (also called nullclines) of autophagy (green) and apoptosis (red) inducers (Figure 2b) [105,106]. The balance curves represent how the steady state of the autophagy inducer varies as a function of the apoptosis inducer and vice versa [107]. Where the curves intersect, the systems might come to rest at a steady state, which can be stable or unstable (see black or white dots on Figure 2b) [59,105]. The phase plane of the autophagy and apoptosis inducers is plotted under physiological conditions (panel left), at low (middle panel) and high (panel right) level of ER stress, respectively [59,107,108].

Under physiological conditions, there is only one not too deep valley in the cellular system which refers to a basal autophagic state [109] (see the black dot with a low level of the autophagy inducer and with the absence of the apoptosis inducer in Figure 2b, panel left) [107]. However, at a low level of ER stress, the mutual antagonism between autophagy and apoptosis might lead to bistability [105,106]. Namely, at a low level of ER stress, the balance curves of the autophagy and apoptosis inducers intersect at three points representing a bistable system with two stable steady states separated by an unstable steady state (Figure 2b, middle panel) [107]. This can be thought of as two valleys (one corresponding to autophagy, the other to apoptosis state) separated by a hill. When the system, like a ball, stands on the top of the hill, it depends on the stress signal which stable valley the ball will “roll into” and stay in [110,111]. A stable state with a high level of autophagy inducer and a low level of apoptotic inducer activity corresponds to autophagy, while a stable state with higher apoptosis inducer activity but lower autophagy inducer activity corresponds to apoptosis [107]. Since the system is bistable, it can theoretically enter any stable state (i.e., autophagy or apoptosis), but since the system starts from a baseline of zero, it enters the autophagic state at low ER stress, while apoptosis remains inactive (see the grey dashed arrow in Figure 2b, middle panel) [107].

With the drastic increase of ER stress, the balance curve of the autophagy inducer does not move, but the balance curve of the apoptosis inducer shifts to the right resulting in the loss of the autophagic steady state (Figure 2b, panel right) [59,107,108]. In this case, the autophagic valley disappears, and the ball representing the system “rolls over” into the only stable state, which corresponds to apoptotic cell death [107], namely, the regulatory system goes to the sole remaining stable state, which corresponds to apoptosis [107]. Interestingly, first, the system tries to induce autophagy by moving towards the “original” autophagic steady state, but in its absence, the outcome is cellular suicide (see the grey dashed arrow in Figure 2b, panel right) [107]. This result suggests that autophagy might precede apoptosis even at a high level of ER stress [105].

### 4.2. Autophagy Always Precedes Apoptosis upon ER Stress

Most of the experimental results follow in time the activity of key components involved in the ER-stress-induced life-and-death decision process [41,112,113]. To demonstrate that the simple model presented below adequately describes the dynamic behaviour of ER-stress-induced UPR, Holczer et al. performed an analysis where they monitored the relative activity of key proteins in time under different levels of ER stress, both experimentally and by computer simulations [105,106,107] (Figure 2c).

Corresponding to experimental results at a low level of ER stress, autophagy is induced in a sigmoid way, while apoptosis remains inactive (Figure 2c, panel left) [105,106,107]. Although UPR tries to turn on apoptosis, the high level of autophagy keeps it inactive due to the double negative feedback between the two mechanisms [105,106,107]. Ogata et al. also confirmed that autophagy becomes induced for cellular survival upon tolerable level of ER stress [41].

In case of an excessive level of ER stress, autophagy has only a transient activity peak; however, later it becomes diminished, while apoptosis turns on (Figure 2c, middle panel) [105]. The autophagy inducer quickly turns on, but the increasing amount of the apoptosis inducer promotes its inactivation when ER stress is not tolerable for the cell [105]. During that time window when autophagy is active, the cell has a chance to save itself. However, the activation of apoptosis is irreversible, and the cell commits suicide [105,106,107]. These results further confirm that the induction of autophagy-dependent survival always precedes the irreversible apoptotic cell death upon ER stress [105,106,107]. This theoretical analysis was experimentally proven with many research groups by using various human cell lines and ER stressors [41,105,114,115,116].

Xu et al. developed an automated live microscope and image analysis to prove that autophagy precedes apoptosis [117]. While autophagy has a transient activation in case of tunicamycin treatment, later, apoptosis is induced in an all-or-none bimodal fashion [117].

Since autophagy stimulates cell survival, and a stronger autophagic response might be able to delay apoptosis, several experiments have been carried out to investigate what happens when cells are pretreated with autophagy inducers [76,105,118]. It has been recently shown that insulin secretion deficiency in beta cells causes ER-stress-mediated cell death [119]. While inhibition of autophagy increases cell death [120,121], rapamycin (an mTOR inhibitor) treatment induced autophagy-promoted cell survival [107,122]. In Kapuy’s lab, various natural agents (such as resveratrol, EGCG, sulforaphane) have been successfully used in recent years to delay apoptotic cell death by autophagy induction with respect to excessive levels of ER stress [76,108,123]. Holczer et al. have also demonstrated by computer simulation that pretreatment with various enhancers of autophagy greatly delays apoptotic cell death, even in the presence of intolerable ER stress (Figure 2c, panel right) [76,108,123].

## 5. Crosslinks Inside and between the UPR Arms and Their Roles upon ER Stress

Novel results have revealed that the three branches of UPR are not just “one-way” signal transduction pathways, and they are not independent from each other but are linked by many cross-links, both within and between branches [124,125], and also with other signal transduction networks, such as the NF-κB pathway [126]. Depending on the sign of the relations, the resulting control loops can be negative, positive or double negative [102]. While positive and double negative feedback loops can cause bistability in the control network, a negative feedback loop can lead to adaptation or even oscillatory kinetics [102] (Figure 3).

Recently, two negative feedback loops have been identified inside the PERK branch [127]. Namely, GADD34 is a regulatory subunit of PP1 phosphatase [128] and is able to dephosphorylate eiF2α [129] (see number “1” connections on Figure 3), assuming that the eiF2α -> ATF4 -> GADD34 −| eiF2α loop enhances the adaptation of the cell to permanent ER stress [130]. By using systems biological methods, Marton et al. have revealed that CHOP inhibits ATF4 upon ER stress (see number “2” connections on Figure 3), supposing that CHOP blocks the hyperactivation of ATF4 upon an excessive level of ER stress [127]. The authors assume that the reduction in the acute ER stress response mechanism is controlled mainly by the ATF4 -> CHOP−| ATF4 feedback loop, while eiF2α -> GADD34 −| eIF2α has an important role in permanent ER stress [127]. Trusina et al. claim that the negative feedback loop in the control network generates a so-called translation attenuation mechanism to the UPR, which has a tighter response upon ER stress [131]. Due to this negative feedback loop, fewer amounts of unfolded protein already ensure the cellular adaptation to stress with less excess chaperone protein [131].

In addition, two positive feedback loops have already been identified within the PERK pathway [127]. Guanabenz-dependent inhibition of GADD34 followed by ER stress induction resulted in a significant decrease of ATF4 protein level, supposing that GADD34 acts positively on ATF4 [127]. This connection establishes the ATF4 -> GADD34 -> ATF4 positive feedback loop in the control network (see number “3” connections on Figure 3) [127]. It has also been shown that CHOP had a positive effect on GADD34 [132,133,134] upon ER stress, resulting in another positive feedback loop, i.e., ATF4 -> CHOP -> GADD34 -> ATF4 (see number “4” connections on Figure 3) [127].

However, positive relationships have been shown not only within the PERK branch, but also between branches upon ER stress [126]. Recent experimental data have revealed that PERK and IRE1 pathways are not independent from each other; rather, they are connected with regulatory loops upon ER stress [90]. Deegan et al. have shown that the inhibition of IRE1 causes the downregulation of CHOP with respect to ER stress, suggesting that the IRE1 pathway has a positive effect on the apoptosis inducer of the PERK pathway [90,135]. In addition, Marton et al. have demonstrated that PERK silencing decreases the phosphorylation state of JNK, thereby validating a PERK-dependent positive effect on the IRE1 arm with respect to ER stress [136]. Additionally, it has also been shown that ATF4 has a positive effect on IRE1 [137,138], while the active, cleaved form of XBP1 is able to induce the activity of both ATF4 and CHOP (see number “5” connections on Figure 3) [139]. These results confirm the presence of IRE1 -> PERK -> IRE1 positive feedback loop at different levels of the given UPR branches.

ATF6 is able to induce both the PERK and IRE1 pathways of UPR with respect to ER stress [140,141,142]. Recent experimental data have suggested that CHOP induction is largely dependent on ATF6 in liver cells during the unfolded protein response [140]. Computational simulations has also been performed to confirm that ATF6 is essential for the proper dynamics of CHOP induction upon ER stress (see the black line of number “6” connections on Figure 3) [141]. Yoshida et al. have proved that the induction of XBP1 mRNA is positively controlled by ATF6 [142]. The spliced active form of XBP1 is observed after the production of the active, nuclear form of ATF6 upon ER stress (see the black line of number “7” connections on Figure 3) [142]. Whether IRE1 and PERK arms are able to enhance the activity of ATF6 pathway is not known yet (see the grey lines of number “6” and “7” connections on Figure 3).

## 6. Discussion

The proper balance of secreted and membrane proteins is controlled in the endoplasmic reticulum (ER), while accumulation of misfolded proteins due to different ER stress events leads to the activation of unfolded protein response (UPR) [22]. The primary role of UPR is to reduce the bulk of damages and try to drive back the system to the former or a new homeostatic state by autophagy-dependent cell survival. However, an excessive level of ER stress results in apoptotic cell death [2,60,143]. According to experimental data, all the three branches of UPR (i.e., PERK, IRE1 and ATF6 pathways) can induce both autophagy and apoptosis (Figure 1) [39,43].

Interestingly, each UPR pathway has members that are able to promote only autophagy, while they have components that can induce both autophagy and apoptosis, as well (Figure 1). The ER-stress-induced dynamic behaviour of the cellular system shows that a low level of ER stress results in a sigmoid induction of autophagy, while apoptosis remains inactive [59,107]. In addition, a transient autophagic survival always precedes apoptotic cell death, even at a high level of ER stress (Figure 2) [105,107]. Therefore, UPR members that only enhance autophagy (such as sXBP1, GADD34) may play a key role in the early stress response (Figure 1), enhancing the survival mechanism. Depletion of either sXBP1 or GADD34 promotes the harmful effect of the ER stress response and turns on apoptosis even at much lower ER stress [82,144,145,146]. Since these regulators are not completely independent of the other members of the UPR (e.g., some results suggest that sXBP1 also induces apoptosis via CHOP [142,147]), it is important to investigate whether they are indeed only capable of inducing autophagy. It is much more likely that they can also induce apoptosis, but to a much lesser extent. They are certainly the key regulators of autophagy in early ER stress response, but they may also be an “emergency reserve” activator of the cell death mechanism. More interesting are those UPR members that have been shown to activate both autophagy and apoptosis (see JNK and CHOP on Figure 1). We claim that these proteins act as a double-edged sword in the ER stress response mechanism, as they play an important role in both survival and cell death processes. We assume that these UPR members are essential in fine-tuning the stress response, depending on the level of ER stress, and may therefore be important therapeutic targets.

Here, it is important to note that it is always the given human disease that determines whether the cells should be “encouraged” to autophagy-dependent survival or rather to die, depending on what is better for the human body. There are diseases where the point is to delay cell death via autophagy induction (e.g., in case of neurodegeneration diseases [148,149]), while in other cases, cell death is more beneficial for the survival of the organism (e.g., during cancer treatment [150]). In these cases, it is very important to decide that these UPR members, which can also induce autophagy and apoptosis, are better to be inhibited or activated during the given treatment. For example, CHOP depletion in β cells provides a therapeutic strategy to alleviate ER stress and dysregulated insulin secretion and consequent fatty liver disease [151]. However, hyper-activation of CHOP with a novel CHOP activator (called LGH00168) significantly suppresses tumour growth; therefore, it seems to be a potential therapeutic agent for human lung cancer [152].

A systems biology analysis of the regulatory control network shows that different regulatory loops envelop the UPR, both between branches and within branches, all of which have a function in the ER stress response (Figure 3) [127,136]. Since hyperactivation of the ER stress response can also be dangerous for the cell (e.g., uncontrolled autophagy can cause cell death [153]), negative feedback loops in the system via eiF2a -> GADD34 −| eiF2α and ATF4 -> CHOP −| ATF4 are very important to prevent over-expression of the UPR members by the target proteins blocking their own activators [127]. Some experimental data have revealed that the negative feedback loop can cause the oscillatory characteristic of autophagy induction upon various stress events (such as rapamycin treatment or food deprivation), thus ensuring that the system can re-use the components generated by self-digestion, but not upon ER stress [154,155]. In a recent theoretical analysis, Erguler et al. explore the dynamical characteristic of the ER stress response mechanism at various levels of stress, and they claim that the intermediate state of UPR, due to moderate level of ER stress, might generate sustained oscillation of the key components of the network [156]. Therefore, it would be worthwhile to investigate experimentally, in the near future, whether this periodic repeat of autophagy can also occur in the case of an intermediate level of ER stress.

More and more results confirm that positive feedback loops are built into the control system, not only within UPR branches (i.e., ATF4 -> GADD34 -> ATF4), but also between them (IRE1 -> ATF4 -> IRE1) (Figure 3) [127,136]. It is well known that the positive-feedback-generated mutual activation is able to create a discontinuous switch of the response as the signal magnitude reaches a critical value [102,103]. Here, we claim these positive feedback loops guarantee the switch-like characteristics of the stress response with respect to ER stress [127,136]. A positive feedback loop may also be good for ensuring irreversible activation of apoptosis in intolerable ER stress [59]. It should be noted here that ATF6 has been shown to have a positive effect on the PERK and IRE1 pathways [140,141,142], but their effect on ATF6 is not yet known. This will have to be clarified experimentally in the future.

Redundant feedback loops in the control network ensure that the response remains robust upon ER stress, even if one or two proteins have some defects [102]. This, however, requires that the dynamic behaviour of UPR proteins that induce autophagy and apoptosis upon ER stress be elucidated in the future in the most precise way.

## 7. Conclusions and Future Directions

Choosing between life and death by the ER-stress-induced UPR is one of the most important tasks of the cells building up an organism. The cellular system has to be capable of adapting to different levels of a tolerable amount of ER stress, such that the process is controlled by so-called autophagy by self-eating the damaged cell compartments. However, an excessive amount of ER stress results in apoptotic cell death to avoid fatal errors. Exploring the kinetic behaviour of the branches of the ER-stress-generated UPR can give us an opportunity to understand the dynamical characteristic of the control network. A generic view from the control network of UPR by using systems biological analysis would be an essential step in the investigation of the regulatory system. Finding possible targets for therapeutic intervention in the case of ER stress has been receiving increasing attention in recent years; therefore, these studies have long-term significance for their medical purposes.

## Figures and Tables

**Figure 1 ijms-25-04368-f001:**
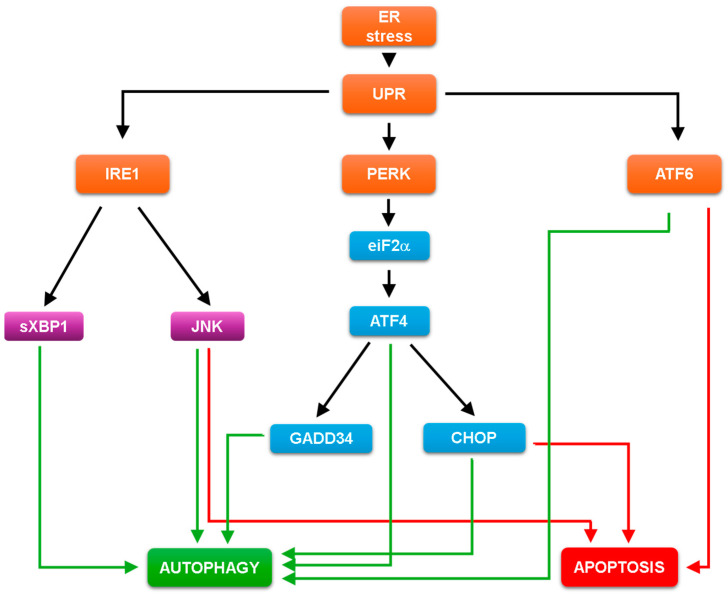
The wiring diagram of UPR with respect to ER stress. Black and coloured continuous lines show how the components can influence each other. While the green arrows represent positive effects on autophagy, the red arrows represent positive effects on apoptosis.

**Figure 2 ijms-25-04368-f002:**
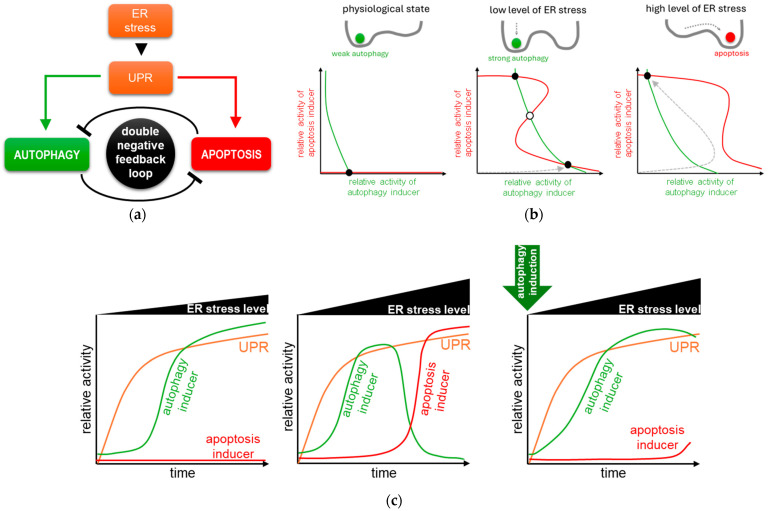
The dynamical characteristic of life-and-death decision with respect to ER stress. (**a**) The simple wiring diagram of autophagy and apoptosis regulation with respect to ER stress. The continuous line shows how the components can influence each other, while blocked end lines denote inhibition. The green arrows represent positive effects on autophagy, while the red arrows represent positive effects on apoptosis. The phase plane analysis of ER stress response (**b**, panel left) under physiological conditions, with respect to (**b**, middle panel) low and (**b**, panel right) high level of ER stress. The balance curves of autophagy inducer (green) and apoptosis inducer (red) are plotted. Trajectories are depicted with grey lines showing which stable state the system is moving towards during a given treatment. The stable and unstable steady states are visualized with black and white dots, respectively. The time series analysis of ER stress response with respect to (**c**, panel left) low and (**c**, middle panel) high level of ER stress and (**c**, panel right) when high level of ER stress is pre-treated with autophagy inducer (green arrow indicates when the autophagy inducer has been added to the system). The relative activity of UPR, autophagy inducer and apoptosis inducer is shown.

**Figure 3 ijms-25-04368-f003:**
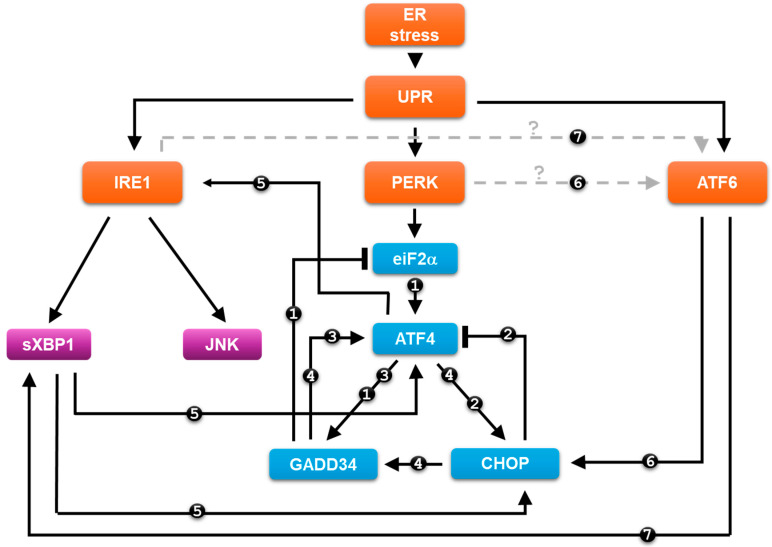
The wiring diagram of crosslinks within and between UPR arms with respect to ER stress. Continuous black line shows how the components can influence each other, while blocked end lines denote inhibition. Dashed grey lines with question mark represent the assumed connections between the regulatory components. The numbers represent the sub-connections of the different positive and negative feedback loops of the control network; the same numbers belonging to the same loop (see text for details).

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
