# Peer review of "Mechanism of Decision Making between Autophagy and Apoptosis Induction upon Endoplasmic Reticulum Stress"

_ijms, 2024, doi:10.3390/ijms25084368_

Round 1

Reviewer 1 Report

Comments and Suggestions for Authors

The manuscript reviews the role of endoplasmatic reticulum stress via the activation of unfolded protein response (UPR) into cellular decision between life and death.

The manuscript is readable but the concept of ER stress must be better described and discussed in the introduction and paragraph 2., before reporting the stress response mechanisms.

More recent literature and studies (2020-2024) must be cited and discussed in the manuscript.

Author Response

"The manuscript reviews the role of endoplasmatic reticulum stress via the activation of unfolded protein response (UPR) into cellular decision between life and death.

The manuscript is readable but the concept of ER stress must be better described and discussed in the introduction and paragraph 2., before reporting the stress response mechanisms."

> I thank the nice comments. I agree with the reviewer that the concept of ER stress was not properly defined. Therefore, the text has been thoroughly revised and extended with a more detailed description about ER stress and its effects (see the highlighted text in the Introduction and Chapter 2).

"More recent literature and studies (2020-2024) must be cited and discussed in the manuscript."

> I am grateful for the reviewer for the useful suggestions. The most recent literature already available has been thoroughly reviewed to find even more relevant publications. The text has been carefully read and revised accordingly.

Reviewer 2 Report

Comments and Suggestions for Authors

In this review article, the author summarized the recent knowledge of the dynamical characteristic of endoplasmic reticulum stress response mechanism by using both theoretical and molecular biological techniques, and payed special attention to describing the mechanism of action of the dynamical features of the feed-back loops controlling cellular life-and-death decision upon ER stress.

Comments

This is an interesting review article. The reviewer has some concerns as follows:

1.     In the end of Introduction, the aim(s) for this review article can be described.

2.     In Figure 1, the dashed or dotted lines generally refer to uncertain signaling pathways, changing them to black continuous lines may be less misunderstood. In addition, the spelling of “eif2alpha” within the figure can be consistent with “eiF2alpha” in the text.

3.     In line 118, “the X-box binding protein 1 (sXBP1)” changes to “the spliced X-box binding protein 1 (sXBP1)”.

4.     In Figure 2c, there is some confusion. In legend, it described that the “(c, panel right)” when excessive level of ER stress is pretreated with autophagy inducer. But the icon for ER stress level (intensity) does not show the difference compared to “(c, middle panel)”. It can be revised.

5.     In Figure 3, the black dashed lines can be changed to black continuous lines to reduce the possibility of misinterpretation. Moreover, the meanings for the numbers within the figure can be explained in the legend.

6.     In lines 282, 287, 372, and 373, there are some confusions. Please check and correct/revise the words for “GADD34 -I eiF2α”, “CHOP -I ATF4”, “GADD34 -I eIF2α‐P”, “GADD34 -I eiF2a”, and “CHOP -I ATF4”, respectively.

7.     In line 270, “NF-kB” changes to “NF-κB” (kappa).

8.     In line 372, “eiF2a” changes to “eiF2α” (alpha).

Author Response

"In this review article, the author summarized the recent knowledge of the dynamical characteristic of endoplasmic reticulum stress response mechanism by using both theoretical and molecular biological techniques, and payed special attention to describing the mechanism of action of the dynamical features of the feed-back loops controlling cellular life-and-death decision upon ER stress.

Comments

This is an interesting review article."

> I thank the nice comments of the reviewer.

"The reviewer has some concerns as follows:

In the end of Introduction, the aim(s) for this review article can be described."

> I am grateful for the reviewer for the useful suggestions. The last paragraph of the Introduction has been extended with the aims of the review.

"In Figure 1, the dashed or dotted lines generally refer to uncertain signaling pathways, changing them to black continuous lines may be less misunderstood. In addition, the spelling of “eif2alpha” within the figure can be consistent with “eiF2alpha” in the text."

> I agree with the reviewer that dashed lines might refer to uncertain signalling pathways, therefore the regulatory connections have been replaced with black or colourful continuous lines. According to this modification the figure legend has been revised as well. The “eif2alpha” on Figure 1 has been changed to “eiF2α” (alpha).

"In line 118, “the X-box binding protein 1 (sXBP1)” changes to “the spliced X-box binding protein 1 (sXBP1)”."

> The “the X-box binding protein 1 (sXBP1)” has been changed to “the spliced X-box binding protein 1 (sXBP1)”.

"In Figure 2c, there is some confusion. In legend, it described that the “(c, panel right)” when excessive level of ER stress is pre-treated with autophagy inducer. But the icon for ER stress level (intensity) does not show the difference compared to “(c, middle panel)”. It can be revised."

> I agree with the reviewer that there is some confusion in the legend. The middle and right panels of Figure 2c have the same stress level, only in the right panel the system was treated with an autophagy inducer before the high stress was induced. Therefore, the icon for ER stress intensity is the same on both figures. As the text was not clear, the Figure legend has been modified.

"In Figure 3, the black dashed lines can be changed to black continuous lines to reduce the possibility of misinterpretation. Moreover, the meanings for the numbers within the figure can be explained in the legend."

> I thank the reviewer the comment. Similarly, to Figure 1 the regulatory connections have been replaced with black or colourful continuous lines. The figure legend has been revised and extended with more details about the meanings for the numbers.

"In lines 282, 287, 372, and 373, there are some confusions. Please check and correct/revise the words for “GADD34 -I eiF2α”, “CHOP -I ATF4”, “GADD34 -I eIF2α‐P”, “GADD34 -I eiF2a”, and “CHOP -I ATF4”, respectively."

> The text has been thoroughly revised and the signs of the connections have been corrected.

"In line 270, “NF-kB” changes to “NF-κB” (kappa)."

> The “NF-kB” has been changed to “NF-κB” (kappa).

"In line 372, “eiF2a” changes to “eiF2α” (alpha)."

> The eiF2a” has been changed to “eiF2α” (alpha).

Reviewer 3 Report

Comments and Suggestions for Authors

 In this review we summarized the recent knowledge of the dynamical characteristic of endoplasmic reticulum stress response mechanism by using both theoretical and molecular biological techniques.

The topic is interesting, and the narrative review well organized.

Please modify the title to make it more clear.

Also, please acknowledge the narrative nature of this review.

I only have a few minor concerns. 

Fig 2 and 3 should be detailed in a clearer way.

Comments on the Quality of English Language

 grammar and syntax errors.

Author Response

"Comments and Suggestions for Authors

In this review we summarized the recent knowledge of the dynamical characteristic of endoplasmic reticulum stress response mechanism by using both theoretical and molecular biological techniques. The topic is interesting, and the narrative review well organized."

> I thank the nice comments of the reviewer.

"Please modify the title to make it more clear."

> I agree with the reviewer that the title of the manuscript was not adequate, there it has been changed to “Mechanism of decision making between autophagy and apoptosis induction upon endoplasmic reticulum stress”.

"Also, please acknowledge the narrative nature of this review.

I only have a few minor concerns.

Comments on the Quality of English Language - grammar and syntax errors."

> I thank the reviewer for pointing out the narrative nature of this review. I have thoroughly revised the whole text, and corrected the grammatical and syntax errors.

"Fig 2 and 3 should be detailed in a clearer way."

> I agree with the reviewer that the figures can be improved. Therefore, the Figure legend of Figure 2 and 3 have been thoroughly revised, and the text has been extended to detail the figures in a much clearer way.

Round 2

Reviewer 2 Report

Comments and Suggestions for Authors

This revised manuscript has a great improvement and can be accepted.